# Effect of internal surface structure of the north wall on Chinese solar greenhouse thermal microclimate based on computational fluid dynamics

**Xingan Liu[1], He Li[2], Yiming Li[2], Xiang Yue[2], Subo Tian[2], Tianlai Li[1]***

**1** College of Horticulture, Shenyang Agricultural University, Shenyang, China, **2** College of Engineering, Shenyang Agricultural University, Shenyang, China

* lxa10157@syau.edu.cn

**Data Availability Statement:** All relevant data are within the manuscript and its Supporting Information files.

## Abstract

Chinese solar greenhouses are unique facility agriculture buildings and widely used in northeastern China, providing a favorable requirement for crop growth. The north wall configurations play an essential role in heat storage and thermal insulation and directly affect the management of the internal environment. This research is devoted to further improve the thermal performance of the greenhouse and explore the potential of the north wall. A mathematical model was designed to investigate the concave-convex wall configurations based on computational fluid dynamics. Four passive heat-storage north walls were analyzed by using the same constituent materials, including a plane wall, a vertical wall, a horizontal wall and an alveolate wall. The numerical model was validated by experimental measurements. The temperature distributions of the north walls were examined and a comparative analysis of the heat storage-release capabilities was carried out. The results showed that the heat-storage capacity of the north wall is affected by the surface structure. Moreover, the critical factor influencing the air temperature is the sum of the heat load released by the wall and the energy increment of greenhouse air. The results suggested that the alveolate wall has preferable thermal accumulation capacity. The concave-convex wall configurations have a wider range of heat transfer performance along the thickness direction, while the plane wall has a superior thermal environment. This study provides a basic theoretical reference to rationally design the internal surface structures of the north wall.

## Introduction

Chinese solar greenhouses (CSG) are unique facility agriculture buildings which widely used in northeastern China. It can provide a favorable environment for crop growth under severe external climate conditions so as to achieve annual production [1–3]. The CSG has the remarkable advantages in term of passive energy storage instead of active auxiliary heating so it can maintain the suitable thermal environment, and its dominating heat source comes from

**Funding:** The present work is financially supported by the Postdoctoral Foundation of Xingan Liu under the Grant 1105/770218004. Xingan Liu had a role in the study design, data collection and analysis, decision to publish, or preparation of the manuscript.

**Competing interests:** The authors have declared that no competing interests exist.

solar radiation [4]. Therefore, the efficient utilization and storage of the received solar energy have great impact on the greenhouse microclimate and can be maximized by controlling the structure parameters of the north wall in order to provide an optimum thermal environment [5]. The north wall configurations play an essential role in heat storage and thermal insulation and directly affect the management of internal environment [6]. Generally, the north wall consists of two parts. The first is the heat storage layer which contributes to large thermal masses are stored in receiving solar radiation during the daytime and release heat into indoor air during the nighttime. The second is a thermal insulation layer which prevents heat loss to the external environment all the time [7]. Therefore, arranging the internal surface structures of the north wall will have significantly impacts on the appropriate climate condition for crop growth in the greenhouse.

Frequently, to improve the heat-storage capability, the material and thickness of the north wall have been ameliorated [8,9]. Guan et al. [10] studied the dynamic heat transfer characteristics of the north wall to determine the appropriate thickness of the three-layer wall consisted of the phase-change material layer. Zhang et al. [11] reported the economic and environmental evaluations of straw block walls and examined the effect of the heat transfer characteristics of north wall on energy distribution. Ren et al. [12] developed a new straw block applied in the north wall to estimate the hygrothermal and fungistatic properties of solar greenhouse. Zhang et al. [13] presented five passive heat storage north walls to increase agriculture productivity in non-arable lands. A lot of studies have shown that the optimization of north wall has strong fundamentally significant for the automation and intelligent control of thermal environment [14,15].

It is necessary to determine the suitable geometry parameters of the north wall due to the variability of solar radiation and uncontrollable external factors [16]. Wang et al. [17] investigated the thermal properties of three north walls with different structure of solar greenhouse based on a simulation analysis method. It was demonstrated that the thermal resistance and thermal inertia impact the minimum air temperature. In another research, Wang et al. [18] improved heat-storage capability by filling soil and perlite in the cavities of a hollow concrete block wall. He et al. [19] mentioned that the removable north wall and the natural ventilation can be achieved through vent dimension arrangement to ensure crop production and quality in greenhouse cultivation. A 2-D transient numerical model was conducted to accurately explore indoor temperature and airflow distribution. For residential buildings with intermittent heating. Wang et al. [20] used CFD technology to forecast the dynamic thermal characteristics of internal walls and the overall heat load. Several studies have shown that the reasonable wall configuration is conducive to reducing energy consumption and improving crop yield and quality, which will be maximized in the utilization benefit of clean energy [11,17]. However, few investigations have been made on the concave-convex wall configurations. In recent years, farmers have attempted to arrange the surface structure of the north wall empirically, so as to increase the air contact area between internal ambiance and wall surface. Unreasonable north walls have been observed to be not effective and the waste of financial and material resources is undesirable. Therefore, additional researches are needed to be performed under the consistent external conditions.

Computational fluid dynamics (CFD) methodology has been used in the present study to provide detailed information about thermal distributions [21–23]. Especially, it can provide a moderate environment and avoid the influence of unstable factors caused by field measurement [24]. In addition, CFD is an effective technique for adequately predicting greenhouse microclimate because it has reliable results and low cost [19,25]. Piscia et al. [26] used a coupling approach to analyze the nocturnal greenhouse microclimate. The results showed that the integration of CFD and energy balance simulations have the ability to provide an accurate prediction. Wang et al [27] simulated the greenhouse microclimate in a typical plastic greenhouse. The deliberately CFD model was developed and could be served as an analytical solution for

greenhouse design investigating. Majdoubi et al. [28] investigated the realism of 3-D models to analyze air circulation and microclimate distribution based on numerical simulation. Most of greenhouse thermal models were validated, and the researchers concluded that CFD is a powerful method with high reliability and accuracy. Nevertheless, the temperature distributions of both internal air and the enclosure are basically dynamic due to the dynamic change of solar radiation transmitted into the greenhouse all the time [29]. Therefore, an investigation on temperature distributions by utilizing a periodically changing external environment is necessary.

The internal surface structure is part of the important factors to determine the greenhouse microclimate and enhance the thermal performance of the north wall [30,31]. In order to accurately analyze the impact of wall configurations on the internal environment of the greenhouse, the influence of crops on the heat exchange of the north wall structure was neglected [32]. The objective of this study is to experimentally and numerically investigate the thermal microclimate and heat transfer characteristics of the north walls with different internal surface structures. There are four categories of the wall configuration: plane wall (PW), vertical wall (VW), horizontal wall (HW) and alveolate wall (AW), where the PW as a comparable case to evaluate other forms of thermodynamic performance in the greenhouse. This study provides a basic theoretical reference to rational design the internal surface structure of the north wall and has significance for the structural arrangement.

## Materials and methods

The experimental validation of the simulation model was conducted by comparing the measurement data on the PW. The experiment can be used to analyze the thermal characteristics of the greenhouse microclimate, and it also can provide a verification model for numerical simulation. Moreover, it ensures the calculation credibility and efficient parameters are optimized for setting initial and boundary conditions.

### Experimental arrangement

The experimental greenhouse with the PW which has been used as the control was located at the Horticulture Facility Design & Environmental Control Research Institute of Shenyang Agriculture University in Shenyang, China (latitude: 41.8˚N, longitude: 123.4˚E, altitude: 42 m). The greenhouse consisted of the north wall, north roof, south roof, gables and soil below the greenhouse, which was orientated in the east-west direction (Fig 1). The length of the greenhouse was 60 m, the width was 8 m, and the height was 4 m. The height of the north wall was 2.7 m and the effective thickness was 0.48 m. The north wall of PW was composed of a 0.11 m thermal insulation polystyrene layer and a 0.37 m sintered brick layer form outside to inside. The north roof consisted of wood boards and straw felts wrapped in waterproof polyethylene film to form a waterproof insulation layer. The south roof was covered by a polyolefin film with a 0.12 mm thickness in order to allow solar radiation to penetrate directly into the greenhouse during the daytime (8:30 a.m.-4:00 p.m.). A cotton blanket with 0.04 m thickness was laid over the south roof for thermal insulation during the nighttime (4:00 p.m.-8:30 a.m. day+1). The experiment period in this paper was the representative severe winter season on January 23, 2017. Since only the specific parameters of the north wall were analyzed and simplified the simulation model under the consideration of calculation. There were no crops in the greenhouse and the exchange conditions of greenhouse ventilation and humidity were also not considered.

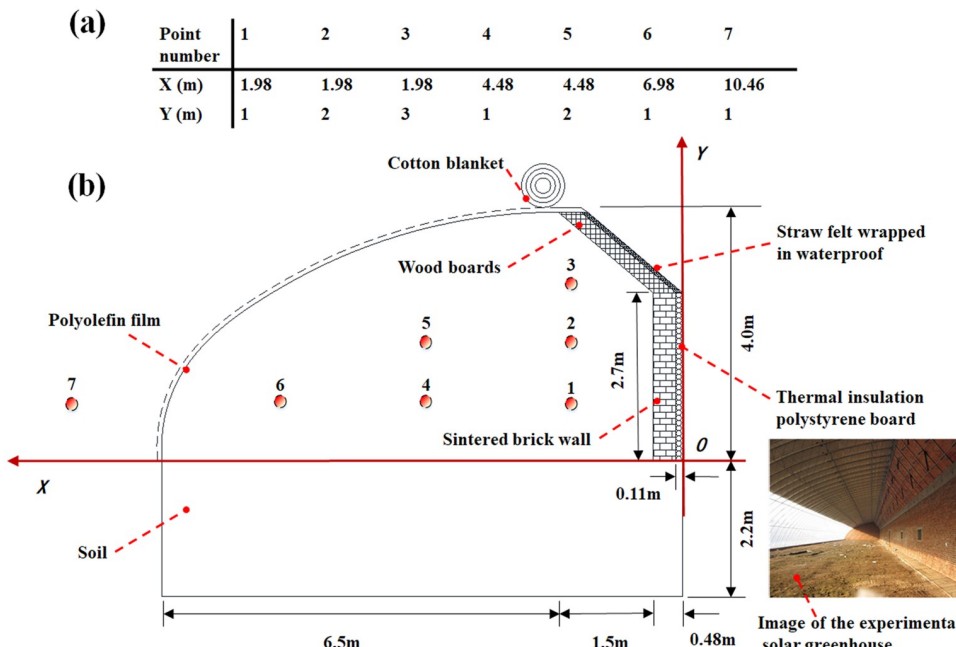

**Fig 1. Schematic of the experimental greenhouse in cross-section view indicating: (a) measurement point coordinates; (b) monitoring locations and component materials for the temperature measurements.**

## Experimental measurement and data collection

For effectively analyzing the greenhouse thermal microclimate, a TYD-ZS2 type environmental data recorder, with instrument temperature measurement accuracy at ±0.1˚C, measuring range -40 to 80˚C, was installed outdoor 1.98 m from the greenhouse to monitor the external temperature so as to provide environment parameters for the simulation model. Meanwhile, as illustrated in Fig 1, six measurement locations were introduced into the greenhouse to obtain quantitative data of the air temperature inside the greenhouse. Air temperature measurement points of indoor and outdoor were 1 through 7. At each experimental point, the temperature was measured at 1-minute intervals and the time-average value was recorded so that a precise temperature equivalent was obtained [33]. All sensors were located in the middle cross-section of the greenhouse and the sensor probes exposed to air would be radiation-proof during the measurement. The data were automatically collected by a recorder (HOBO U100-011, USA Oneset Co Ltd, accuracy at ±0.21˚C, measuring range -20 to 70˚C) and finally output in Excel format.

In order to evaluate temperature distribution of the north wall, the temperatures at different layers of the PW configuration were measured by thermocouples (T-type, USA, accuracy at ±0.1˚C). The data were expediently recorded for 10-minute intervals by CR1000 and CR3000 data collectors. The measurement positions were 8 through 14 (Fig 2).

## Mathematical modelling

**Physical model.** The calculational domain of the greenhouse included the interior air, the heat storage wall consisted of sintered bricks wall and thermal insulation polystyrene layer, the north roof composed of wood boards and straw felts, the soil below the greenhouse and the front roof composed of a polyolefin film during the day and a cotton blanket during the night. The thermo-physical properties of the constituent materials in the greenhouse are shown in

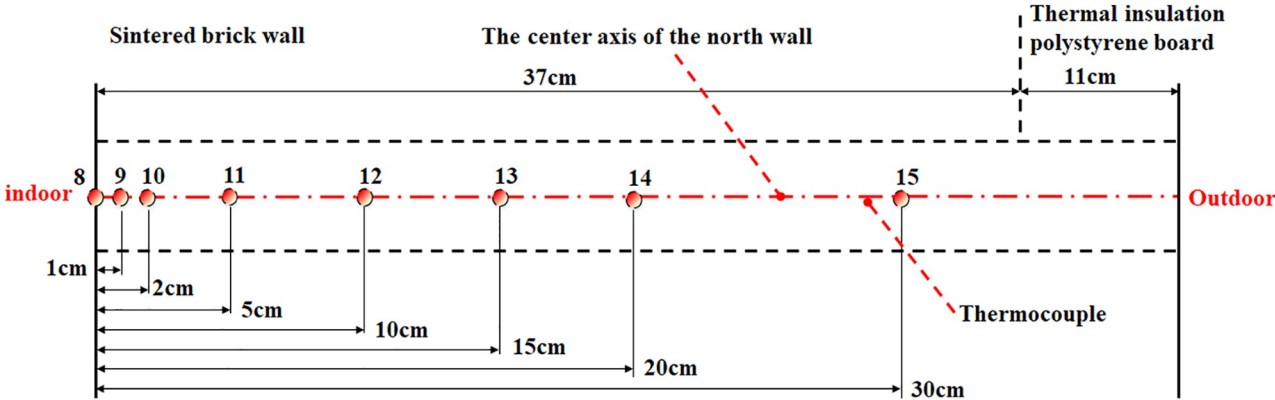

**Fig 2. Sectional view of the measurement points at the horizontal height of 1.35m on the north wall in the center of the greenhouse.**

Table 1 [34]. The concave-convex wall greenhouses had the same geometric parameters as a traditional plane wall (Fig 3a) greenhouse except that the structure shape of the wall surface was different. The north wall of the CCW had three layers from outside to inside: 0.11 m thermal insulation polystyrene layer and 0.28 m sintered bricks wall inside and 0.18 m wall surface structure layer, in which the effective thickness of wall surface structure layer was equivalent to 0.09 m the sintered bricks wall. However, the surface of VW and HW (Fig 3b and 3c) was formed by the sintered brick wall with a thickness of 0.18 m and the air domain with a concave-convex spacing of 0.18 m were staggered and arranged, respectively. The surface layer of AW (Fig 3d) consisted of a square alveolate air domain of 0.18 m and a sintered brick wall of 0.18 m thickness surrounding the air domain. In order to qualitatively analyze the influence of the north wall structure on internal thermal microclimate, the sensible and latent heat exchange caused by humidity changes in the greenhouse was ignored, and greenhouse crops and ventilation conditions were also not considered. Hexahedral mesh was utilized to improve calculation accuracy. Credited simulation results could be obtained when the meshes were controlled at approximately 2.5 million cells and the mesh skewness was below 0.7. When the value of skewness is smaller, the superiority of mesh quality computing is higher, and the value cannot be recommended to exceed 0.9 for simulation [35].

**Governing equations.** CFD technique is a discretized numerical method used to simulate fluid flow and heat transfer in geometric areas [36,37], including three fundamental conservation equations (i.e. mass, momentum and energy equations). The conservation of mass partial differential equation (Eq 1) means that the sum of the net mass of the fluid flowing out of the control body in unit time is equal to the mass decrease of the control body due to density change in the same time interval.

$$\frac{\partial \rho}{\partial t} + \nabla(\rho \vec{v}) = S_m \tag{1}$$

The momentum partial differential equation (Eq 2) can be described as the rate of changing of momentum with respect to time of a given fluid element is equal to the sum of various external forces acting on the element body.

$$\frac{\partial}{\partial t}(\rho \vec{v}) + \nabla(\rho \vec{v}\vec{v}) = -\nabla P + \nabla \cdot (\bar{\bar{\tau}}_{eff}) + \rho \vec{g} + \vec{F} \tag{2}$$

The essence of the energy partial differential equation (Eq 3) is the first law of

**Table 1. Thermo-physical properties of materials used in the simulation.**

| Material | Density (kg m$^{-3}$) | Specific heat capacity (J kg$^{-1}$ °C$^{-1}$) | Thermal conductivity (W m$^{-1}$ °C$^{-1}$) |
|---|---|---|---|
| Internal air | Ideal-gas | 1006.43 | 0.024 |
| Sintered brick | 1600 | 1051.1 | 0.5 |
| Polystyrene board | 30 | 2414.8 | 0.041 |
| Wood board | 550 | 2510 | 0.29 |
| Straw felt | 300 | 1680 | 0.13 |
| Polyolefin film | 950 | 1600 | 0.19 |
| Cotton blanket | 150 | 1880 | 0.06 |
| Soil | 1700 | 1010 | 0.85 |

thermodynamics, which is specifically described as the rate of increase of energy in the element is equal to the net heat flux entering the element gradient plus the work done by mass force and surface force on the element.

$$\frac{\partial}{\partial t}(\rho E) + \nabla \cdot (\vec{v}(\rho E + P)) = \nabla \cdot \left(k_{eff}\nabla T - \sum_j h_j \vec{J}_j + (\bar{\bar{\tau}}_{eff} \cdot \vec{v})\right) + S_h \tag{3}$$

The standard k-ε model (Eqs 4 and 5) is a semi-empirical model applicable to a wide range of turbulent flows. It assumes that turbulence is isotropic and computationally robust.

$$\frac{\partial}{\partial t}(\rho k) + \frac{\partial}{\partial x_i}(\rho k u_i) = \frac{\partial}{\partial x_j}\left[\left(\mu + \frac{\mu_t}{\sigma_k}\right)\frac{\partial k}{\partial x_j}\right] + G_k + G_b - \rho\varepsilon - Y_M + S_k \tag{4}$$

$$\frac{\partial}{\partial t}(\rho\varepsilon) + \frac{\partial}{\partial x_i}(\rho\varepsilon u_i) = \frac{\partial}{\partial x_j}\left[\left(\mu + \frac{\mu_t}{\sigma_\varepsilon}\right)\frac{\partial\varepsilon}{\partial x_j}\right] + C_{1\varepsilon}\frac{\varepsilon}{k}(G_k + G_{3\varepsilon}G_b) - C_{2\varepsilon}\rho\frac{\varepsilon^2}{k} + S_\varepsilon \tag{5}$$

The solar radiation was applied to simulate the solar radiation of complex geometry with CCW configurations using the P-1 Radiation Model (Eq 6) due to reducing the calculate cost

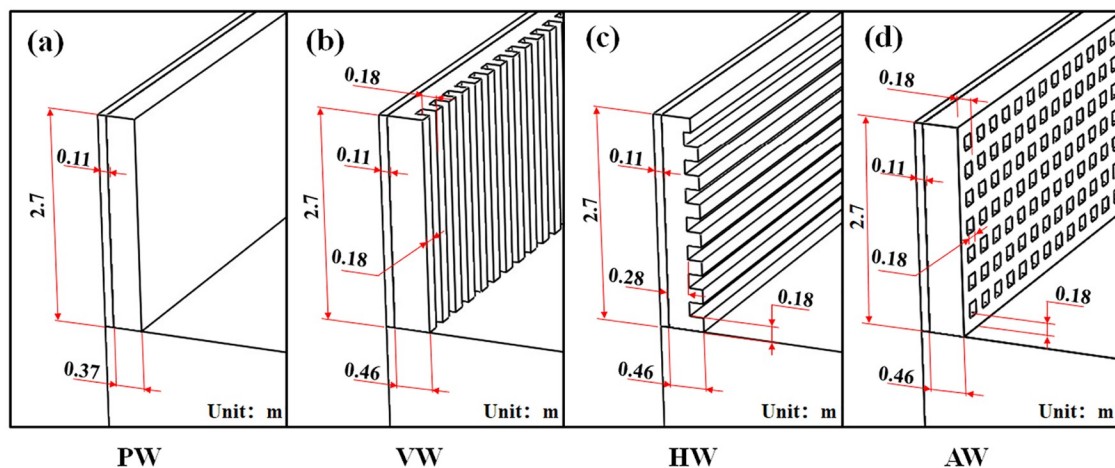

**Fig 3. Structural configuration diagram of the north wall models for the thermal performance analysis.**

only by solving a diffusion equation.

$$q_r = -\frac{1}{3(a + \sigma_s) - C_0\sigma_s} \nabla I \tag{6}$$

The transport equation for $I$ is

$$-\nabla \cdot q_r - aI + 4an^2\sigma T^4 = S_I \tag{7}$$

The expression for $\nabla \cdot q_r$ (Eq 7) can be directly substituted into the energy equation (Eq 3) to account for heat sources due to radiation. Meanwhile, the Solar Ray Tracing Model (Eq 8) was activated. This model is only suitable for 3-D models and is based on ray a tracing algorithm to simulate the dynamic changes of solar radiation.

$$\frac{dI(\vec{r}, \vec{s})}{ds} + (a + \sigma_s)I(\vec{r}, \vec{s}) = an^2\frac{\sigma T^4}{\pi} + \frac{\sigma_s}{4\pi}\int_0^{4\pi} I(\vec{r}, \vec{s}')\Phi(\vec{s} \cdot \vec{s}')d\Omega' \tag{8}$$

Where, geographical location and time zone, model orientation, sunshine factor and simulation time were imported to the model. In order to control the radiation update frequency during continuous phase solution proceeds, the energy iteration per radiation parameter was set to 10 default and illumination parameters enabled solar-calculator.

The Boussinesq model (Eq 9) which has been successfully applied in nature-convection flows, considering the change of airflow caused by the stack effect in the greenhouse. This model assumes that the fluid density is constant in all solved equations except for the buoyancy term in the momentum equation along the direction of gravity.

$$(\rho - \rho_0)g = -\rho_0\beta(T - T_0)g \tag{9}$$

To evaluate the heat storage-release performance of north walls with different configurations, the corresponding energy can be calculated through the average surface temperature difference of disparate depth layers in the wall. The heat storage-release per unit volume of the wall in unit time for the north wall can be written in Eq 10. Total heat storage-release capacities under different wall configurations can be calculated as Eq 11.

$$q_\tau = \rho c \frac{(t_{j+1} - t_j)}{\Delta\tau} \tag{10}$$

$$Q = \sum_{i=1}^{n} V \cdot q_\tau = Q_a + Q_b \tag{11}$$

where $q_\tau$ is the heat absorbed or heat released per unit volume of the wall in unit time, and a positive value means heat absorption, while a negative value means heat release; $\rho$ is the density of the wall configuration; $c$ is the specific heat capacity of the wall configuration; $\Delta\tau$ is the unit volume of the wall; $t_j$ is the surface average temperature at a specified depth $j$; $Q$ is the internal energy of the wall; $V$ is the volume of wall thickness layer in different wall configurations; $Q_a$ is the amount of heat absorption; $Q_b$ is the amount of heat release.

**Numerical details.** The boundary conditions and initial conditions were determined according to the measured data including the external temperature and the patch temperature, Since the microclimate changes dynamically with time, a transient 3-D model was adopted and turbulence was activated using the standard k-ε model with standard wall functions applied to the near-wall treatment [38]. The SIMPLE scheme was employed based on the pressure and velocity coupling solver and second order upwind was adopted for density and energy

[39]. The absolute criteria of residual were $10^{-6}$ for the energy equation and $10^{-3}$ for mass and momentum equations, and $10^{-4}$ for others. After the residual value was not reduced (for the total residual of all variables in each equation), the iterative step size was finally determined to maximal iterate 20 times for each time step size [40]. In order to obtain more accurate simulated data, one case was automatically saved every 10 minutes in real-time during the calculation process. The experiment period was calculated repeatedly to avoid the influence of path temperature on the thermal microclimate, and finally, consistent parameters were obtained as experimental comparison data.

## Results and discussion

In order to investigate the terminal performance of the CSG with different wall configurations, parameter researches of the north wall thermal properties and the internal environment have been carried out. Utilizing the CFD simulation software, an improved numerical model of a modified wall has been developed and validated by the experiment results.

### Validation of the simulation model

The simulation model was validated by comparing simulation results with the experiment results in the same environmental conditions. The time-average temperature of each measurement point (P1-P6) was taken as the measurement results, and compared with the simulated average air temperature. Fig 4a demonstrates that air average temperature predicted by simulation for every measurement point is in good agreement with experiment results. For example, the average absolute temperature difference between measurement and simulation was 0.22˚C. The average relative error between the measurements and simulation results of the air temperature was about 9.5%. The simulation results were somewhat below measurement value during the daytime due to the thermocouple shells were heated by solar radiation and have certain heat preservation function. Subsequently, the simulation results were slightly higher measurement value after the thermally insulated quilt was expanded, and the reason for this is that the thermally insulated quilt was actually somewhat colder than the simulation model [8]. In order to analyze the temperature distribution, the average temperatures of the north wall are compared with the experimental data in Fig 4b. The average relative error between the measurement results and simulation results of the wall temperature was about 4.2%, with a maximum error of about 7.1% at point 15 and a minimum error of about 2.7% at point 13. It indicates that the average interior temperature distribution is reasonable to some extent,

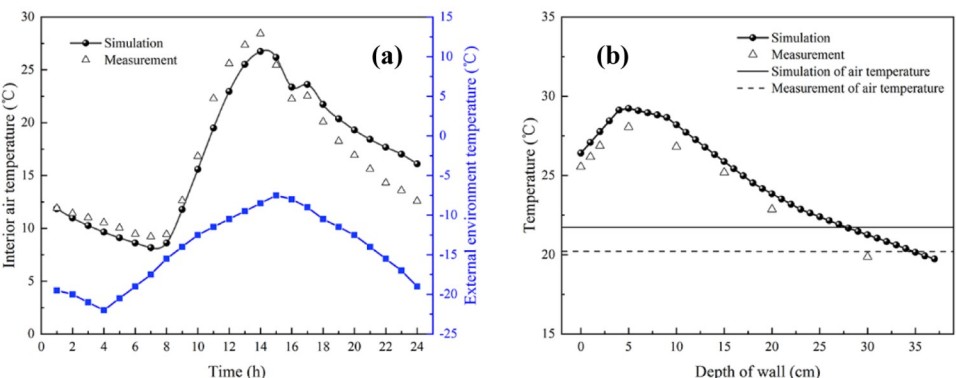

**Fig 4. Comparison of simulation and experimental data.** (a) Indoor Average temperature; (b) The average temperatures of the north wall.

which further increases the credibility of simulation. Base on the above results, the unaltered boundary conditions, solver and material parameters were implemented in all cases so as to conveniently predict the effect of different internal surface structure of the north wall on the thermal microclimate of the CSG.

## Characteristics of greenhouse microclimate change

The indoor air is enclosed by various maintenance structures to form an approximately closed thermal growth environment. The variation trends of the surface average temperature of different structures were similar patterns (Fig 5). In the representative PW, the temperature difference between the north wall surface and the wood board surface was basically maintained at 5.3°C, while the average temperature difference between the soil surface and the wood board surface during the night was 1.3°C higher than that during the day. For the north roof composed of wood board and straw felt, there was insufficient heat accumulation capacity for the whole day. Therefore, its main function is thermal insulation. The plastic film of the south roof was influenced by the external cold airflow and easily caused the greenhouse microclimate fluctuation, suggesting that the south roof is the primary source of greenhouse energy loss. The surface temperature of the north wall was always slightly higher than that of the soil surface, and the surface temperature difference reached a peak of 5.8°C at 2:00 p.m. Furthermore, the average temperature difference between the internal wall surface and the soil surface during the day was 2.9°C higher than that during the night. It is suggested that the heat storage-release performance of the north wall is obviously higher than that of the soil, and the north wall plays a decisive role in the thermal environment of the CSG.

Fig 6 shows the temperature distributions on the central cross-section of the greenhouse along the east-west direction at 7:00 a.m., 2:00 p.m. and 0:00 a.m. The outside temperature began to rise with the increase of the external solar radiation and the insulated quilt was rolled up at 8:30 a.m. to accommodate the photosynthesis of crops. The air temperature reached the maximum value at 2:00 p.m. and the inner surface temperatures of the wall and soil increased

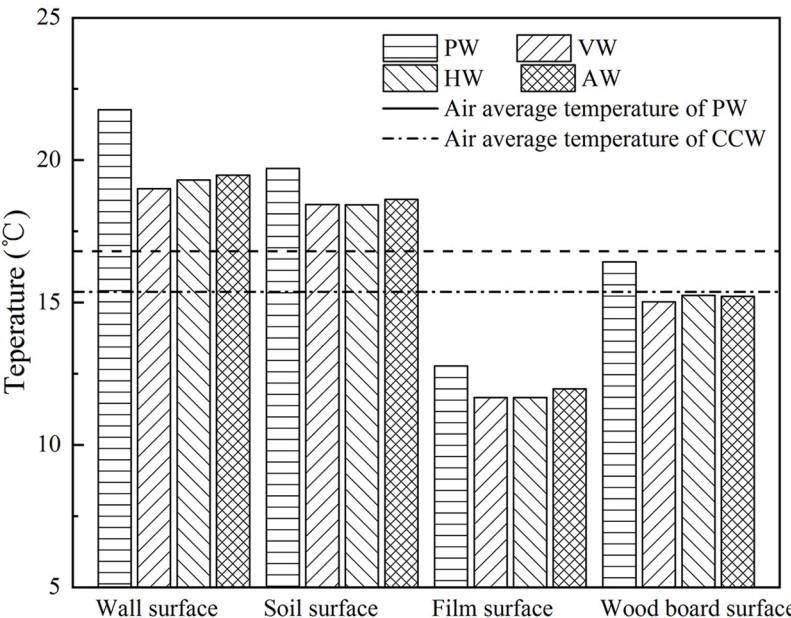

**Fig 5. The surface average temperature of internal air maintenance structures in the whole day.**

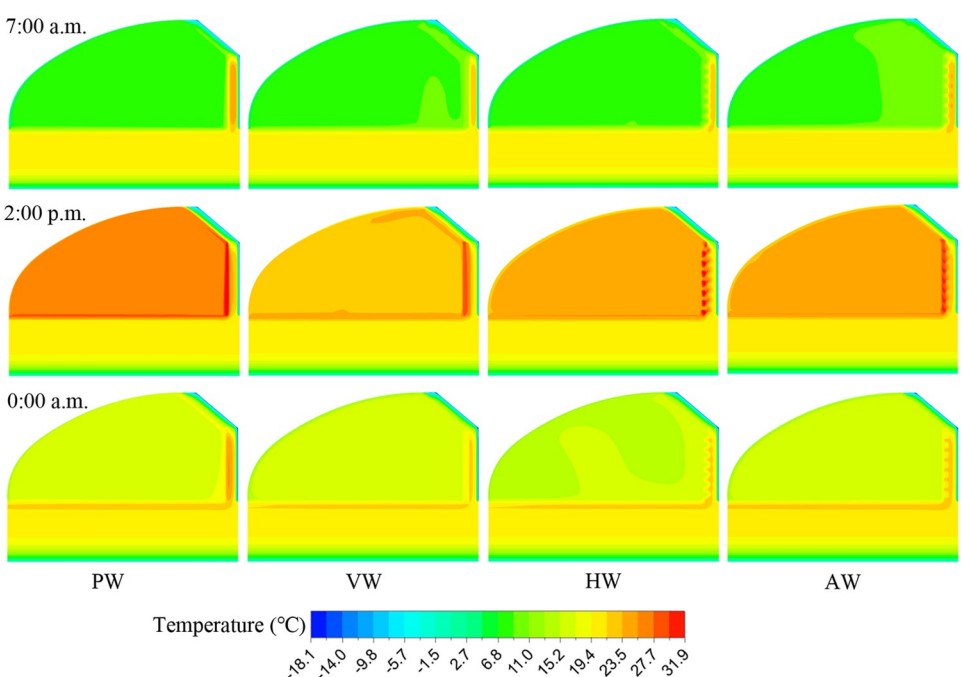

**Fig 6. Temperature distributions for the four wall configurations at 7:00 a.m., 2:00 p.m. and 0:00 a.m. on the central cross-sections of the CSG.**

significantly due to the radiant heat transfer of the internal air in the meantime. The insulated quilt was urgently needed to be unfolded at 4:00 p.m. to provide a satisfactory growing environment when the external solar intensity was insufficient and the external temperature began to drop [8]. The wall and soil of the greenhouse gradually released heat into the internal air by convection heat transfer to resist the energy loss from the south roof during the nighttime. Due to the limited solar radiation energy received on the north wall surface, the energy increment of the air temperature was indirectly transferred to the north wall for accumulation, resulting in that the air temperature of CCW was lower than that of PW during the daytime. Moreover, the north wall could continuously transfer heat to the greenhouse after the insulated quilt was unfolded. However, the energy would be partially attenuated after the heat was accumulated through the wall and then released into the greenhouse, where the attenuated part was used to maintain the temperature of the north wall. It could be concluded that the critical factor influencing the air temperature is the sum of the heat load released by the wall and the energy increment of greenhouse air. The energy increment represents an unconverted heat load that does not pass through the wall.

Internal air temperature variation inside the greenhouse with time for different wall configurations in the whole day is shown in Fig 7a. The average air temperature of PW greenhouse was 14.9˚C when the insulated quilt was expanded. VW and AW greenhouses had the approximate average air temperature, which was 0.9˚C lower than that PW greenhouse. While the lowest average air temperature of HW greenhouse was 13.6˚C. The results suggested that the greenhouse with PW has a superior thermal environment in general. The air temperature rose sharply after the insulated quilt was unfolded. The highest air temperature difference between greenhouses with different wall configurations appeared at 2:00 p.m. Meanwhile, the air temperatures of both north walls also reached their peaks. The highest air temperature of PW greenhouse was 26.7˚C, which was higher than HW (i.e. 3.1˚C), AW (i.e. 3.3˚C) and VW (i.e. 3.6˚C) greenhouses, respectively. The significant decrease in air temperature after 2:00 p.m.

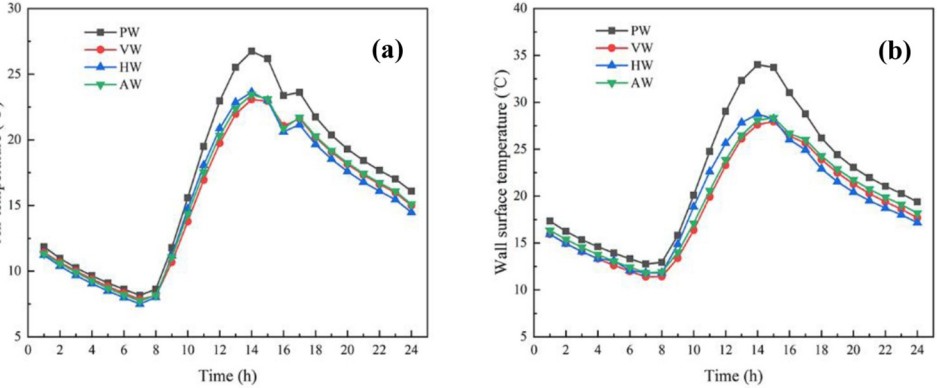

**Fig 7. Temperature variation for four wall configurations.** (a) The internal air temperature; (b) The internal surface temperature of the north walls with different configurations.

was attributed to the temperature reduction of the ambient environment. Since the insulated quilt was unfurled to avert the energy loss from the south roof, the insulated measure also slightly enhanced the air temperature within the next hour due to the heat release from the walls and soil is higher than the heat dissipation from the south roof, thereafter the air temperature tended to decrease steadily due to the attenuation of heat release. The smallest temperature fluctuation value appeared in the VW greenhouse, with the specific data of 15.2˚C. It indicates that the thermal stability of the VW greenhouse is better than that of other greenhouses with different configurations. The time history variation of the internal surface temperature of different wall configurations is reflected in Fig 7b. The results showed that the interior surface temperature of PW was higher than that of CCW at all time periods. The peak time of the internal surface temperature of HW and AW (i.e. 3:00 p.m.) was one hour later than that of PW and HW. Moreover, the highest average temperature rose rate of PW was 3 ˚C/h from the time when the insulation was rolled up to the highest temperature, which was higher than that of HW (i.e. 0.6 ˚C/h), VW (i.e. 0.95 ˚C/h) and AW (i.e. 0.96 ˚C/h), respectively. It indicated that the heat-storage capacity of CCW is higher than that of PW, but the improving air temperature is much lower than that of PW. In addition, the transverse structure of HW obstructs the airflow along the wall caused by thermal pressure, resulting in the average heating rate slightly higher than that of VW and AW.

The plane temperature was defined as the average temperature of certain soil layer. As shown in Fig 8. The amplitude of temperature difference was significantly affected by wall configuration when the depth of the soil was between 0 cm and 50 cm at 0:00 a.m. Nevertheless, the average soil temperature chiefly maintains an approximate value of 19.8˚C when the soil depth was 50 cm. The reason for stability was that there were thermal insulation boards around the soil to prevent the heat loss from the greenhouse to outside. The maximum temperature difference between PW and CCW was 1.2˚C at the uniform depth. The soil temperature continued to elevate with the increase of the depth in the range of 0~15 cm, while the soil temperature decreased gently when the soil depth was 15~30 cm. It could be concluded that the soil under the depth of 30 cm is the major source of heat transfer to the internal air.

## Temperature distribution and heat transfer of north walls

Fig 9 shows the temperature fluctuation conditions of the different north wall configurations. In the temperature contours of VW and AW configurations, (a) means the convex part of

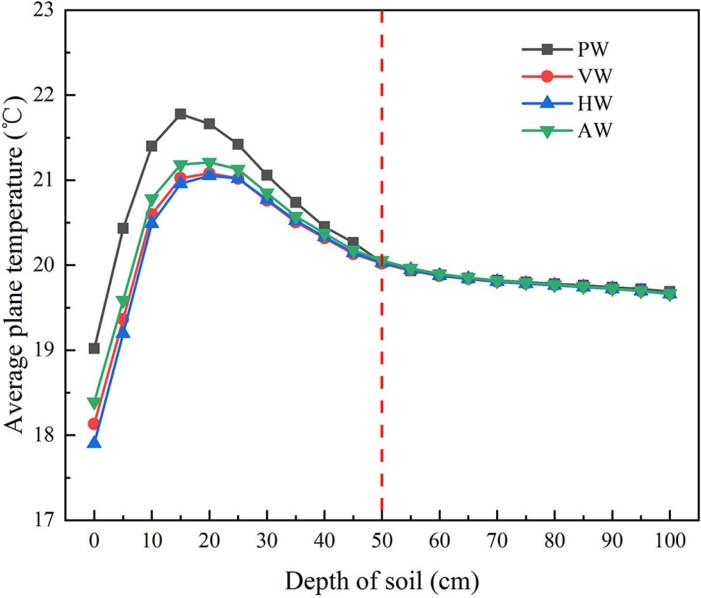

**Fig 8. The average plane temperature of the soil layer below the greenhouse at 0:00 a.m.**

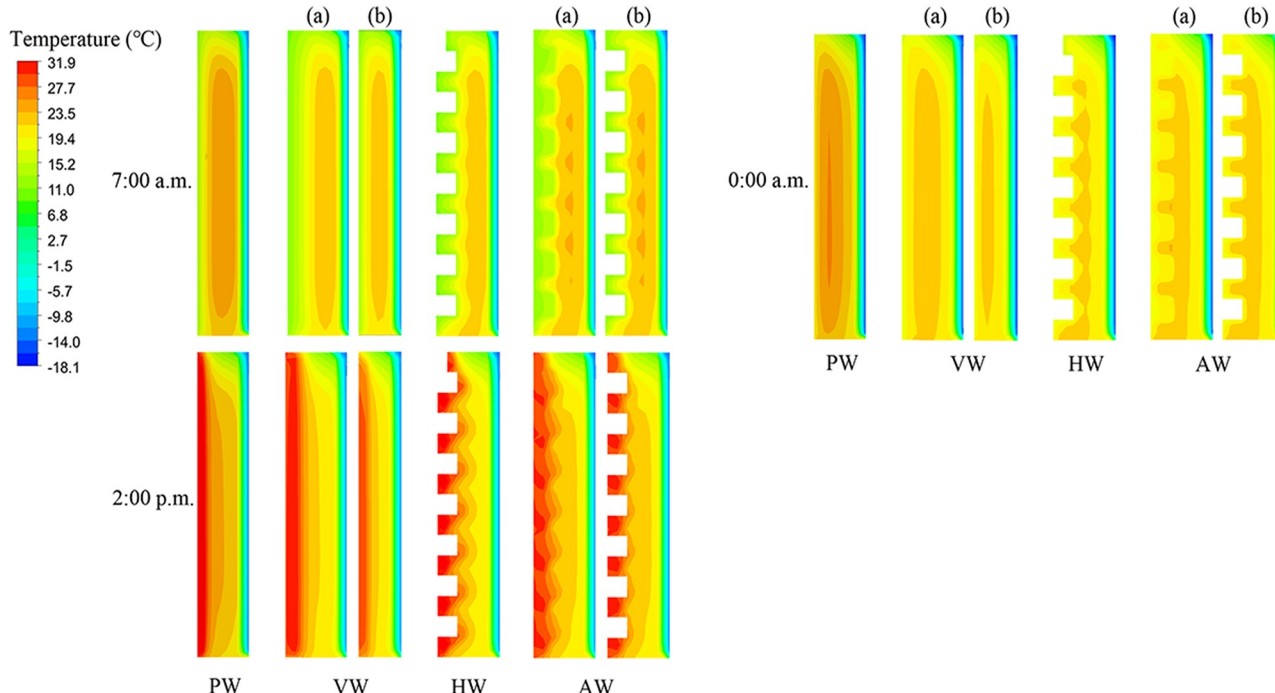

**Fig 9. Temperature distributions of four north wall configurations at 7:00 a.m., 2:00 p.m. and 0:00 a.m. on cross-sections of the CSG.** (a) The central cross-section of the convex position is equivalent to the central cross-section of the CSG; (b) The central cross-section of a concave position is adjacent to the convex part.

walls, and (b) represents the concave part of walls. The 3-D unsteady heat transfer process of the north wall is influenced by the structure and solar radiation environment. It was noticeable that the highest interior surface temperature of the north walls appeared at 2:00 p.m. (the boundary area on the left side of the wall was internal air), and the temperature distributions of the north walls were discrepant due to the different wall configurations. The air contact area of VW and HW was 307.8 m$^2$, which was 1.9 times of PW. The air contact area of AW was 318 m$^2$. It could be observed that the temperature distribution of the north wall is affected by air contact area, suggesting that the heat storage-release performance of the north wall is determined by the wall structure.

It can be seen intuitively from Fig 9 that the thermal mass of PW was depicted as a rectangular shape in the temperature distribution, whereas the CCW configurations appeared as respective wavy shapes with the discrepant structure of the interior surface. It indicated that the temperature distribution in PW is more effectively concentrated than that in the CCW. Moreover, the north wall of the greenhouse plays an important role in heat storage during the daytime and heat release during the nighttime. The obtained results are consistent with that of [41].

In order to evaluate the temperature change in the north walls with different configurations, the hourly temperature distributions of the depth are drawn as described in Fig 10. The results showed that the temperature fluctuation of PW was 13.1~34.8°C, while VW, HW and AW were 10.6~32°C, 10.1~31°C and 10.8~29.4°C, respectively. It was concluded that the temperature difference of PW was 0.3°C and 0.8°C higher than that of VW and HW, what was remarkable was that 3.1°C higher than that of AW. In addition, the maximum internal temperature of

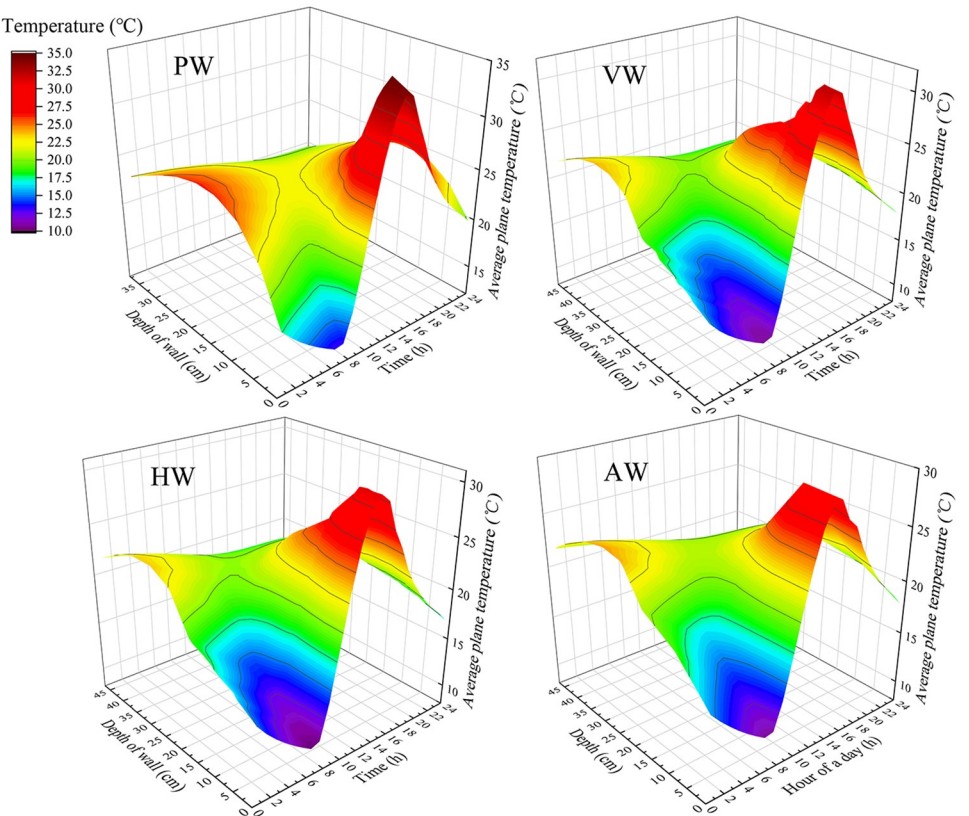

**Fig 10. Hourly temperature distributions of different north wall configurations.**

AW was 5.4°C, 2.6°C and 1.6°C lower than that of PW, VW and HW, respectively. The reason was the effective volume of AW was 7.8 m$^3$ larger than that of other north walls with different configurations. For the same equivalent volume, the maximum temperature of VW and HW was lower than that of PW at 2.8°C and 3.8°C.

The temperature fluctuation was defined as the temperature difference between the maximum and minimum at a certain layer. Fig 11 shows that the wall depth of 0~18 cm is the wall structural layer of CCW and 18~46 cm is the sintered brick wall of CCW, where the dotted line represents the interior surface of PW. For concave-convex wall structures, the temperature fluctuation range of AW in the wall structure layer with 9 cm thickness was 16~18.3°C, and it was 1.0~3.6°C lower than that of VW and HW. The average temperature fluctuation of VW in the interval of 0~5 cm was 0.5°C higher than that of HW. It indicated that the thermal stability of AW is obviously superior to CCW on the wall structure layer. In addition, in order to analyze the heat transfer characteristics, the temperature decay rate was defined as the rate of temperature decline in the north wall thickness. The temperature decay rate of AW in the range of 0~9 cm was 0.29 °C/cm lower than that in the range of 9~22 cm was 0.54 °C/cm, and the temperature fluctuation after 23~46 cm basically tended to be stable and maintains at 3.6°C. The temperature decay rate of VW and HW in the range of 0~5 cm was 0.59 °C/cm lower than that in the range of 5~25 cm. For the traditional plane structure wall, the temperature decay rate of PW was 0.56 °C/cm in the range of 9~14 cm, and the temperature decay rate was below 0.47 °C/cm in the range of 14~25 cm. The temperature decay rate of the wall temperature remains basically stable after 25 cm [41]. It showed that the wall attenuation is divided into three layers: namely delay attenuation layer, rapid attenuation layer and stable attenuation layer.

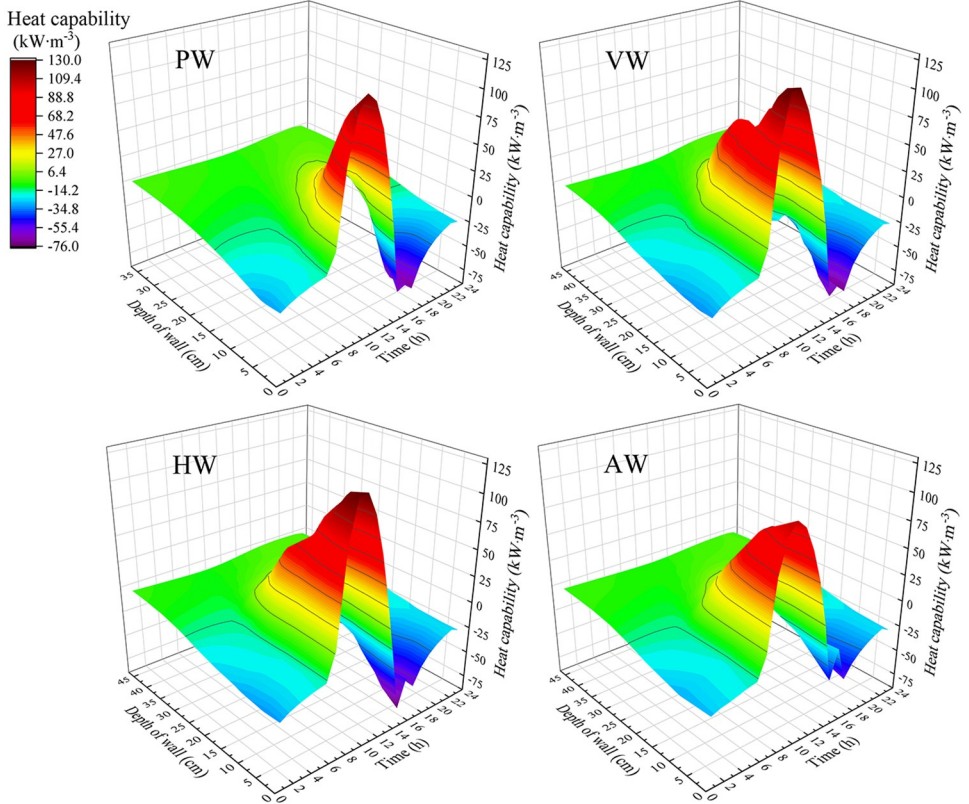

**Fig 11. The temperature fluctuation of different north wall configurations at specific depths.**

### Heat storage-release characteristics of north walls

Hourly heat storage-release performance distributions of the north walls with different configurations under unit volume are shown in Fig 12. On the one hand, the CCW had a wider range of heat storage-release along the thickness direction than PW. On the other hand, the heat storage period was delayed with the decrease of heat storage depth and the heat storage amount of the north wall decreased rapidly, which indicated the wall structure adjustment is an important method to improve the heat storage performance of the CSG. However, the increase of wall depth cannot improve the total amount of heat storage, and the heat storage capacity of the north wall is affected by the surface configuration.

Fig 13a illustrates the relationship between wall depth and heat storage capacity. Obviously, it could be seen that the heat storage capacity of the wall decreased with the increase of wall depth. The heat storage energy of AW basically approached to 0 when the wall depth was 35 cm, while the heat storage depths of VW and HW were 40 cm and 39 cm, respectively. It indicated that the heat storage depth of VW was superior. The heat storage depth of PW could reach 30 cm. It indicated that the heat storage depth of CCW is greater than that of PW. Additionally, HW was always 12.7 kW/m$^3$ higher than VW in the sintered brick wall layer due to the transverse structure of HW impeded the interior airflow upward along the wall surface, resulting in local airflow stagnation and higher thermal accumulation. The comparison between PW and CCW configurations showed that the average heat storage performance of PW within the wall depth of 9~15 cm was approximately 77 kW/m$^3$ higher than that of VW and HW, and 115.5 kW/m$^3$ higher than that of AW. However, in the subsequent 16~39 cm interval, the heat storage performance of PW was 35.6 kW/m$^3$ lower than that of VW and HW. What's more, the effective volume of AW was 7.8 m$^3$ higher than that of other wall configurations, which was mainly concentrated in the wall structure layer. Fig 13b shows the

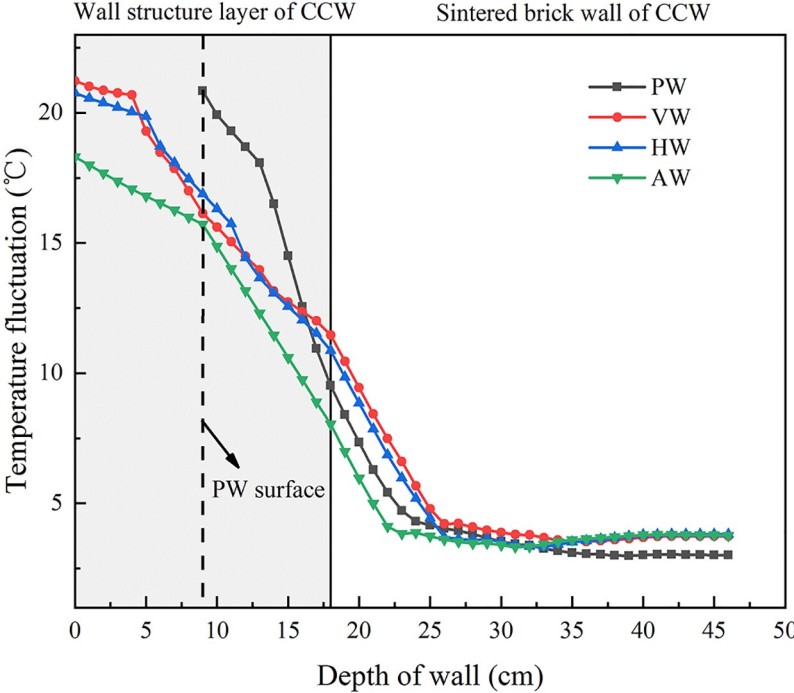

**Fig 12. Hourly heat capability distribution covers both heat storage and release of different wall configurations.**

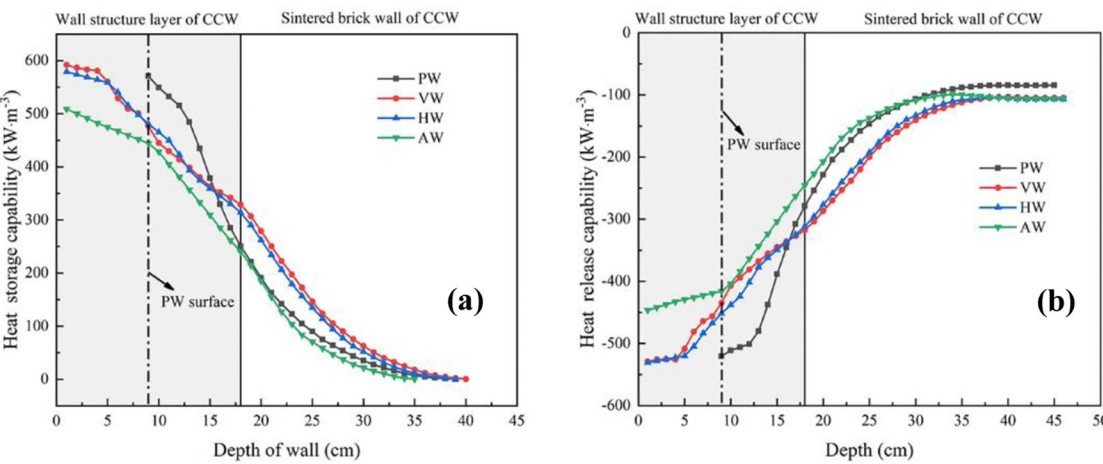

**Fig 13. Heat transfer characteristics of different depth layers for the north walls.** (a) Heat storage capabilities; (b) Heat release capabilities.

variations of the heat release capacities with the wall depth. It clearly indicated that heat release capacities of different wall configurations were positively correlated with the heat storage capacities and the depth of heat release corresponds to the depth of heat accumulation. However, the heat release capacity of PW was 19.9 kW/m$^3$ slightly lower than that of CCW, which indicated that CCW transmits additional energy to the external 0.11 m polyethylene insulation board than PW. This phenomenon was caused by insufficient thermal preservation that the heat storage depth of CCW was 4~5 cm more than that of PW. Therefore, the thermal preservation requirements of CCW have an urgent require to be improved by enhancing the external insulation thickness.

As shown in Fig 14, the total heat storage-release capacities of different wall configurations were obtained by numerical simulation with the identical external environment and the

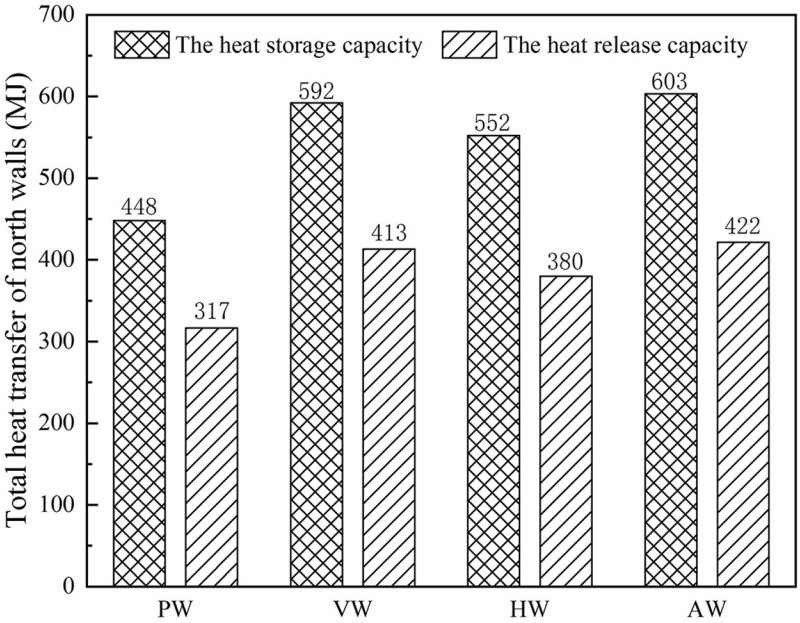

**Fig 14. Total heat storage-release capacities under different wall configurations.**

specified greenhouse geometry dimension. According to Eq 11, the total heat storage volume was larger than the total heat release volume. The difference values of heat storage-release capacities of the four wall configurations were 131.7 MJ, 179 MJ, 172.2 MJ and 181.7 MJ, respectively. The excessive energy was stored in several walls with different configurations to maintain the heat storage body temperature. The heat storage capacity of VW and HW was 32.2% and 23.2% higher than that of PW with equivalent volume. During the heat storage stage from 8:30 a.m. to 4:00 p.m., the heat storage capacity of AW was 34.6% higher than that of PW. During the heat release stage from 4:00 p.m. to 0:00 a.m., the heat release capacity of AW was 33.1% higher than that of PW. It indicated that the CCW has a better heat transfer performance than the PW.

## Conclusion

In present research, the conventional PW was evaluated to explain heat transfer characteristics of CCW configurations including vertical, horizontal and alveolate thermal structures applying the computational fluid dynamics method. The accuracy of simulation was validated by experiment results. According to the simulation results, the south roof of the greenhouse is the main source of energy loss when the north roof basically no heat accumulation capacity and its main function is heat preservation and thermal insulation. Moreover, the heat storage-release performance of the north wall is obviously higher than that of the soil, suggesting that the north wall plays a decisive role in the thermal environment of the CSG. In addition, the greenhouse soil depth of approximately 30 cm is the major part of heat transfer to the air for different wall configurations. The temperature distribution in the traditional PW is more effectively concentrated than that in the CCW, and the critical factor influencing the air temperature is the sum of the heat load released by the wall and the energy increment of greenhouse air. The energy increment represents an unconverted heat load that does not pass through. What' more, the greenhouse with traditional PW has a superior thermal environment in general than that with other structural configurations. The reason is the limited solar radiation energy on the north wall surface. The energy of elevated air temperature is indirectly transferred to the north wall for accumulation, resulting in the air temperature of CCW is lower than that of PW during the daytime, subsequently, the north wall can continuously transfer heat to the greenhouse after the insulated quilt is unfolded. However, the energy will be partially attenuated after the heat is accumulated through the wall and then release into the greenhouse, where the attenuated part is used to maintain the temperature of the north wall. AW has further thermal accumulation capacity because the effective volume of AW is 7.8 $m^3$ larger than that of other walls in the wall structure layer. Nevertheless, for the same equivalent volume, CCW has a wider range of heat storage-release along the thickness direction than PW. The wall structure is an important adjustment method to improve the heat storage performance of the greenhouse. Therefore, the increase of wall depth cannot improve the total amount of heat storage, and the heat storage-release capacity of the north wall is affected by the surface structure.

## Supporting information

**S1 Table. Nomenclature.**
(DOCX)

## Acknowledgments

The authors are grateful to Horticulture Facility Design & Environmental Control Research Institute for supporting the project.

## Author Contributions

**Conceptualization:** Xingan Liu, He Li, Tianlai Li.

**Data curation:** Xingan Liu, He Li.

**Formal analysis:** Xingan Liu, Yiming Li.

**Funding acquisition:** Xingan Liu.

**Investigation:** He Li, Xiang Yue, Subo Tian, Tianlai Li.

**Methodology:** He Li, Yiming Li.

**Project administration:** Xiang Yue.

**Resources:** Xiang Yue, Subo Tian, Tianlai Li.

**Software:** Yiming Li.

**Supervision:** Subo Tian.

**Validation:** Xingan Liu, Xiang Yue.

**Writing – original draft:** Xingan Liu, He Li, Yiming Li.

**Writing – review & editing:** Tianlai Li.

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
