## [Decision Letter · Decision Letter 0]

17 Mar 2020

PONE-D-20-05091

Effect of internal surface structure of the north wall on Chinese solar greenhouse thermal microclimate based on computational fluid dynamics

PLOS ONE

Dear Dr Li,

Thank you for submitting your manuscript to PLOS ONE. After careful consideration, we feel that it has merit but does not fully meet PLOS ONE’s publication criteria as it currently stands. Therefore, we invite you to submit a revised version of the manuscript that addresses the points raised during the review process.

We would appreciate receiving your revised manuscript by May 01 2020 11:59PM. To enhance the reproducibility of your results, we recommend that if applicable you deposit your laboratory protocols in protocols.io, where a protocol can be assigned its own identifier (DOI) such that it can be cited independently in the future. For instructions see: http://journals.plos.org/plosone/s/submission-guidelines#loc-laboratory-protocols

We look forward to receiving your revised manuscript.

Kind regards,

Djamel Eddine Ameziani, Prof

Academic Editor

PLOS ONE

Journal Requirements:

Reviewers' comments:

Reviewer's Responses to Questions

**Comments to the Author**

1. Is the manuscript technically sound, and do the data support the conclusions?

Reviewer #1: Yes

Reviewer #2: Partly

2. Has the statistical analysis been performed appropriately and rigorously? 

Reviewer #1: No

Reviewer #2: N/A

3. Have the authors made all data underlying the findings in their manuscript fully available?

Reviewer #1: Yes

Reviewer #2: No

4. Is the manuscript presented in an intelligible fashion and written in standard English?

Reviewer #1: Yes

Reviewer #2: Yes

5. Review Comments to the Author

Reviewer #1: The following questions should be considered:

1. What is the advancement of the present CFD simulation model compared with those in published references?

2. What is the reason that makes the time difference between the simulation and experiment when the lowest value comes in figure 4a?

3. What is the possible change of the concusion if there are crops in the greenhouse?

Reviewer #2: -The introduction is very long

-Under the Boussinesq assumption, there is the thermal/concentration dependency of the density. Why the term mass does not appear

-Figure 2: the position of the center axis of the north wall is not clear, At what value of the y axis.

--Authors should present the relative error between numerical and experimental results.

The authors should explain how they calculated the mean temperature numerically and experimentally.

-How to explain the increase in temperature between 16h00 and 17h00 (Fig. 4 and Fig. 7)

-The manuscript has many grammatical errors. Please eliminate all grammatical errors. Please take this advice seriously.

6. PLOS authors have the option to publish the peer review history of their article (what does this mean?). If published, this will include your full peer review and any attached files.

Reviewer #1: Yes: Yali Guo

Reviewer #2: No

---

## [Author Response · Author response to Decision Letter 0]

19 Mar 2020

Dear editor and reviewers:

The authors would like to acknowledge the encouraging and valuable comments concerning our manuscript entitled “Effect of internal surface structure of the north wall on Chinese solar greenhouse thermal microclimate based on computational fluid dynamics”. (ID: PONE-D-20-05091).Those comments are very helpful for revising and improving our paper, as well as the important guiding significance to our researches. We have studied the comments carefully and made corrections which we hope meet with approval. The main corrections are in the manuscript and the responds to the reviewers’ comments are as follows.

Replies to the reviewers’ comments:

Reviewer #1:

Thanks very much for your kind work and consideration of the publication of our paper. On behalf of my co-authors, we would like to express our great appreciation to the editor and reviewers. Considering the Reviewer’s suggestions, we tried our best to improve the manuscript. All of the inappropriate clarifications have been modified in the revised manuscript.

1. What is the advancement of the present CFD simulation model compared with those in published references?

Response: A mathematical model was designed to investigate the concave-convex wall configurations based on computational fluid dynamics. To the best of our knowledge, there are no published research works that combine the concave-convex wall configurations and CFD simulation. With regard to the application of CFD simulation model of the wall, most researches have focused on the traditional plane wall.

2. What is the reason that makes the time difference between the simulation and experiment when the lowest value comes in figure 4a?

Response: The time difference between simulation and experiment is mainly due to the heat preservation function of the thermocouple shell, and the time difference is more obvious when the thermocouple shell is heated by continuous solar radiation. The above explanation has been included in the revised manuscript (see line 284-289). The relevant literature is cited as follows:

[1] Tong G, Christopher DM, Li B. Numerical modelling of temperature variations in a Chinese solar greenhouse. Comput Electron Agric. 2009;68: 129–139. doi:10.1016/j.compag.2009.05.004

3. What is the possible change of the conclusion if there are crops in the greenhouse?

Response: That’s a very good idea, thank you. The goal of the greenhouse wall structure design is to improve the crop growth environment. Both crop transpiration and photosynthesis will have a certain effect on the greenhouse microclimate environment, and they will attenuate the room temperature fluctuation amplitude of the greenhouse. However, the presence of crops does not completely change the heat transfer relationship between the wall structure and the indoor air. The qualitative consideration of microclimate impacts requires an additional influence factor. Therefore, the coupling relationship between crop thermal characteristics and wall structure types needs to be further studied.

Reviewer #2: -The introduction is very long

Thanks very much for your kind work and consideration on publication of our paper. On behalf of my co-authors, we would like to express our great appreciation to editor and reviewers. Considering the Reviewer’s suggestions, we tried our best to cut out the unnecessary content of the introduction.

1. Under the Boussinesq assumption, there is the thermal/concentration dependency of the density. Why the term mass does not appear.

Response: This model assumes that the fluid density is constant in all solved equations except for the buoyancy term in the momentum equation along the direction of gravity. We refer to the following journal.

[1] Saberian A, Sajadiye SM. The effect of dynamic solar heat load on the greenhouse microclimate using CFD simulation. Renew Energy. 2019;138: 722–737. doi:10.1016/j.renene.2019.01.108

2. Figure 2: the position of the center axis of the north wall is not clear, At what value of the y axis.

Response: The authors agree that the original statement is inappropriate. the wording “in center of the north wall” has been revised as “at the horizontal height of 1.35m on the north wall in the center of the greenhouse”. The above explanation has been included in the revised manuscript (see line 163-164).

3. Authors should present the relative error between numerical and experimental results.

The authors should explain how they calculated the mean temperature numerically and experimentally.

Response: Thank you for the advice, we've listened to your suggestions and made changes throughout the manuscript. The average relative error between the measurements and simulation results of the air temperature was about 9.5%. The average relative error between the measurement results and simulation results of the wall temperature was about 4.2%, with a maximum error of about 7.1% at point 15 and a minimum error of about 2.7% at point 13. The above explanation has been included in the revised manuscript (see line 282-283, 291-293).

4. How to explain the increase in temperature between 16h00 and 17h00 (Fig. 4 and Fig. 7)

Response: The significant decrease in air temperature after 2:00 p.m. was attributed to the temperature reduction of the ambient environment. Since the insulated quilt was unfurled to avert the energy loss from the south roof, the insulated measure also slightly enhanced the air temperature within the next hour due to the heat release from the walls and soil is higher than the heat dissipation from the south roof, thereafter the air temperature tended to decrease steadily due to the attenuation of heat release. The above explanation has been included in the revised manuscript (see line 360-366).

5. The manuscript has many grammatical errors. Please eliminate all grammatical errors. Please take this advice seriously.

Response: We are very sorry for our incorrect writing. We have further checked the grammar mistakes and made amendments that marked in the revised manuscript with track changes.

Once again, thank you very much for your constructive comments and suggestions which would help us both in English and in depth to improve the quality of the paper.

Kind regards,

Xingan Liu

E-mail: 383666179@qq.com

Corresponding author: Tianlai Li

E-mail address: lxa10157@syau.edu.cn

---

## [Editor Report · Decision Letter 1]

23 Mar 2020

Effect of internal surface structure of the north wall on Chinese solar greenhouse thermal microclimate based on computational fluid dynamics

PONE-D-20-05091R1

Dear Dr. Li,

We are pleased to inform you that your manuscript has been judged scientifically suitable for publication and will be formally accepted for publication once it complies with all outstanding technical requirements.

With kind regards,

Djamel Eddine Ameziani, Prof

Academic Editor

PLOS ONE

Additional Editor Comments (optional):

Dear Tianlai Li

Thank you for the manuscript (and the revised one) which you have submitted for possible publication in the PLOS One Journal. I advise you that the The responses to the comments of the reviewers are appropriate. I am pleased to advise you that I intend to accept your work for publication In its current form.

Sincerely yours
---

## [Editor Report · Acceptance letter]

24 Mar 2020

PONE-D-20-05091R1 

Effect of internal surface structure of the north wall on Chinese solar greenhouse thermal microclimate based on computational fluid dynamics 

Dear Dr. Li:

I am pleased to inform you that your manuscript has been deemed suitable for publication in PLOS ONE. Congratulations! Your manuscript is now with our production department. 

With kind regards,

on behalf of

Dr. Djamel Eddine Ameziani 

Academic Editor

PLOS ONE